# Magnetic Resonance Imaging after Nasopharyngeal Endoscopic Resection and Skull Base Reconstruction

**DOI:** 10.3390/jcm13092624

**Published:** 2024-04-29

**Authors:** Paolo Rondi, Marco Ravanelli, Vittorio Rampinelli, Intan Zariza Hussain, Marco Ramanzin, Nunzia Di Meo, Andrea Borghesi, Michele Tomasoni, Alberto Schreiber, Davide Mattavelli, Cesare Piazza, Davide Farina

**Affiliations:** 1Radiology Unit, Department of Medical and Surgical Specialties, Radiological Sciences and Public Health, University of Brescia, Spedali Civili, Piazzale Spedali Civili 1, 25123 Brescia, Italy; paolo.rondi@unibs.it (P.R.); nunziadimeo92@gmail.com (N.D.M.); andrea.borghesi@unibs.it (A.B.); davide.farina@unibs.it (D.F.); 2Otolaryngology Unit, Department of Medical and Surgical Specialties, Radiological Sciences and Public Health, University of Brescia, Spedali Civili, Piazzale Spedali Civili 1, 25123 Brescia, Italy; 3Department of Radiology, Universiti Kebangsaan Malaysia Medical Center, Jalan Yaacob Latiff, Cheras, Kuala Lumpur 56000, Malaysia

**Keywords:** nasopharyngeal endoscopic resection, magnetic resonance, magnetic resonance imaging, skull base, skull base reconstruction, endoscopic surgery, nasoseptal flap, temporo-parietal fascia flap

## Abstract

**Background:** Postoperative imaging after nasopharyngeal endoscopic resection (NER) and skull base reconstruction is quite challenging due to the complexity of the post-surgical and regional anatomy. **Methods:** In this retrospective observational study, we included patients treated with NER from 2009 to 2019 and submitted to Magnetic Resonance Imaging (MRI) 6 and 12 months after surgery. A radiologist with 15 years of experience analyzed all MRI scans. **Results:** A total of 50 patients were considered in this study, 18 of whom were excluded due to imaging unavailability, and 16 of whom were not considered due to major complications and/or persistent disease. Sixteen patients were evaluated to identify the expected findings. Inflammatory changes were observed in 16/64 subsites, and regression of these changes was observed in 8/64 at 1 year. Fibrosis was observed in 5/64 subsites and was unmodified at 1 year. The nasoseptal flap showed homogeneous enhancement at 6 months (100%) and at 1 year. The temporo-parietal fascia flap (TPFF) showed a decrease in the T2- signal intensity of the mucosal layer in 57% of the patients at 1 year and a decrease in enhancement in 43%. **Conclusions:** Identifying the expected findings after NER and skull base reconstruction has a pivotal role in the identification of complications and recurrence.

## 1. Introduction

Nasopharyngeal Carcinoma (NPC) is an overall rare malignancy of the head and neck with age-standardized rates generally below 1 per 100,000. Higher prevalence has been observed in the Cantonese population of southern China (where it is considered endemic), and intermediate rates are found in indigenous populations in Southeast Asia, the Arctic region, North Africa and the Middle East [1].

NPC is a multifactorial disease with genetic, environmental, infective, dietary and occupational factors1; Epstein-Barr virus (EBV) infection is strongly associated with NPC [2].

NPC is histologically classified as keratinizing squamous cell carcinoma (type I); differentiated nonkeratinizing carcinoma (type II); undifferentiated nonkeratinizing carcinoma (type III); or basaloid squamous cell carcinoma.

NPC is highly radio-sensitive; radiotherapy is the elective treatment, with or without concomitant chemotherapy [3]. Surgery is not an option for first-line treatment but might play a role in cases of local and/or regional recurrence.

Since the 1990s, transnasal endoscopic surgery has been progressively and routinely used for the treatment of a wide spectrum of head and neck diseases. More recently, it has been used in the management of neoplastic conditions; this mini-invasive approach has been introduced in the treatment of selected malignancies of the nasopharynx, such as the recurrence or persistence of nasopharyngeal carcinoma [4,5,6]. Endoscopic nasopharyngeal surgery has also been used in selected cases for the management of olfactory neuroblastoma [7,8], adenoid cystic carcinoma [9] and other malignancies.

The introduction of endoscopic nasopharyngeal surgery in the treatment of nasopharyngeal malignancies has reduced morbidity compared to open surgery approaches or re-irradiation; furthermore, increasing surgical expertise, the development of a better anatomic understanding of the skull base anatomy, and technological advances in instrumentation in recent years have contributed to higher survival outcomes [10,11,12].

## 2. Materials and Methods

### 2.1. Objectives

The aim of this study was to analyze expected findings in the skull base and deep spaces of the neck after NER and flap reconstruction.

### 2.2. Patients

We considered consecutive patients treated with NER between January 2009 and January 2019.

The inclusion criterion was the availability of postoperative imaging studies at 6 months and 1 year after NER; the exclusion criteria were persistent disease and major complication after surgery.

For the evaluation of the flap, we included all patients with at least one imaging study after NER, regardless of the presence of recurrent lesions or major complications.

### 2.3. Surgical Procedure

Nasopharyngeal endoscopic resection (NER) is a surgical procedure aimed at treating nasopharyngeal tumors, particularly those that are localized and recurrent after radiation therapy. This technique utilizes endoscopic equipment to access the nasopharynx, exploiting the corridor provided by the nasal cavities, offering a less invasive alternative to traditional open surgery. It significantly reduces patient morbidity, shortens hospital stays and improves the postoperative recovery period, while providing oncologic results not inferior to those of open surgery [13,14]. The multidisciplinary team must consider re-irradiation, surgery and chemotherapy (in combination or not) when dealing with recurrent nasopharyngeal carcinomas [13]. Castelnuovo et al. subdivided NER procedures into three different types [15,16]. Besides the NER type, posterior septectomy can enhance exposure and surgical maneuverability:-Type 1 NER: This type of resection is limited to the postero-superior nasopharyngeal walls, reaching the bony floor of the sphenoid sinus and clivus. The lateral limits of the approach are the torus and the Rosenmüller fossa. The cartilaginous portion of the Eustachian tube is preserved. Posteriorly, the ventral portion of the clivus may be drilled out if needed.-Type 2 NER: Cranially, this type of resection is extended to include the anterior wall, the rostrum, and the floor of the sphenoid sinus. The sphenoidal mucosa is removed and the floor of the sinus is entirely drilled. Laterally, the cartilaginous portion of the Eustachian tube is preserved. Posteriorly, the resection reaches the periosteum of the skull base and the ventral portion of the clivus, which can be drilled if necessary.-Type 3 NER: This approach is extended to include the lateral nasopharyngeal wall up to the parapharyngeal space and the cartilaginous portion of the Eustachian tube. On the midline, this type of surgical resection is similar to type 2 NER; the resection of the nasal septum is combined with a complete ethmoidectomy and medial maxillectomy, with exposure of the posterior maxillary wall. The content of the pterygomaxillary fossa is exposed, and the maxillary artery is identified and clipped. If the nasoseptal flap is harvested, the maxillary artery and peripheral septal branch are preserved. The root of the pterygoid process is drilled. According to the need for lateral exposure, the medial and lateral pterygoid laminae can be removed. When required, the floor of the middle cranial fossa (great wing of the sphenoid) and the lateral foramina (rotundum and ovale) can be reached. This approach allows for the removal of the upper parapharyngeal space. The carotid artery may be exposed.

### 2.4. Skull Base Reconstruction after NER

Reconstruction can follow ablative surgery if defect coverage and promotion of the healing process are needed. Typically, the reconstruction has to provide coverage to the exposed bone to avoid delayed osteomyelitis. Moreover, the carotid artery must be protected from chronic inflammation and potential blowouts. Wide defects require the use of pedicled flaps, which may guarantee an adequate amount of well-vascularized tissue.

The most commonly used pedicle is the Hadad flap, a mucoperiosteum and mucoperichondrial flap based on the nasoseptal artery, the branch of the sphenopalatine artery and the terminal branch of the internal maxillary artery. The use of the Hadad flap is contraindicated when the defect is wide and the quality of the flap is hampered by previous treatments [17].

Alternative mucosal pedicled flaps have been developed, such as the middle and inferior turbinate flaps, to be used in selected cases when the Hadad flap is not available and the defect is limited.

The temporoparietal fascial flap (TPFF) is a regional pedicle flap. It encompasses the harvesting of temporal subcutaneous tissue, the temporoparietal fascia, loose areolar tissue and the temporalis muscle fascia. It is pedicled on the superficial temporal artery, and transposed if the nasal fossa is accessed through a trans-pterygoid route [18].

The advantages of this flap are predictable vascular anatomy, wide dimensions and thickness, a long vascular pedicle and rich vascularization with subsequent quick healing even in unfavorable conditions such as in previously irradiated patients [17,18,19,20].

### 2.5. MR Protocol

MR imaging (MRI) studies were performed on a 1.5T scanner (Magnetom Aera; Siemens, Erlangen, Germany) with a dedicated head and neck 20-channel phased-array coil.

The acquisition protocol was as follows:T2-weighted Turbo Spin Echo sequences on the axial and coronal planes, section thickness = 3 mm, matrix = 512 × 208.T1-weighted Turbo Spin Echo sequence on the axial plane, section thickness = 3 mm, matrix = 512 × 208.A 3D fat-saturated gradient echo sequence with an isotropic spatial resolution of 0.6 mm after gadolinium-based contrast agent injection.Diffusion-weighted imaging sequence, TE = 59 ms, section thickness = 3 mm, matrix = 132 × 132, b-values = 0 and 1000 mm/s^2^. ADC maps were generated automatically.

Multiplanar reconstructions on the three main orthogonal planes were obtained from the 3D post-contrast fat-saturated gradient echo image with a 1.2 mm slice thickness.

### 2.6. Image Analysis

One radiologist (M.R.) with 15 years of experience in head and neck imaging evaluated all the scans, analyzing the soft-tissue subsites and skull base bones from the same side of the NER (for lateralized spaces) and median spaces.

The following criteria were applied in the analysis:

Soft-tissue changes were evaluated in four anatomical subsites: the parapharyngeal space; retropharyngeal/prevertebral space; carotid space; and masticator space.

Skull base bone changes were evaluated in the clivus, the atlo-occipital junction, the second cervical vertebra (C2), the petrous apex and the base of the pterygoid processes.

The inflammation of bone or soft tissues was defined as an area of abnormal signal with no mass effect and the following pattern:T2 hyperintensity, T1 hypointensity and no enhancement after contrast medium administration (heralding edema).T2 intermediate/high signal, T1 hypointensity and enhancement after contrast medium administration (suggesting granulation tissue).

Fibrosis is defined by the presence of hypointense tissue both on T2- and T1-weighted sequences, with mild or absent contrast enhancement.

Bone sclerosis is defined as the hypointensity of bone marrow on T2- and T1-weighted sequences with no enhancement.

The nasoseptal flap was evaluated considering enhancement and thickness changes over time.

The TPFF was subdivided into three layers: superficial/mucosal, intermediate and deep layers; for each layer, we reported the MRI signal (T2 and T1) and enhancement and their changes over time; signal intensity was evaluated by comparing the signal of each layer to that of normal muscle.

### 2.7. Statistical Analysis

Due to the low population size of our cohort, statistical analysis of the data was minimal. Fisher’s exact test and Freeman–Halton test were used to compare the MRI findings between the first and second MRI and between the two different flaps (nasoseptal and TPFF). Statistical significance was fixed at 0.05. The analyses were conducted in MedCalc (MedCalc^®^ Statistical Software version 20, MedCalc Software Ltd., Ostend, Belgium; https://www.medcalc.org; 2021).

## 3. Results

### 3.1. Patients’ Characteristics

Fifty consecutive patients treated with NER between 2009 and October 2019 were considered for this study; 18 patients were excluded since imaging was not available.

Of the remaining 32 patients 16 were not considered for the evaluation of the expected findings due to early major complications and persistent disease; 16 patients were included and considered for the evaluation of expected findings after NER.

Twenty-four out of thirty-two patients required reconstruction with a pedicled flap. The nasoseptal flap was used in 14/24 patients, while TPFF was used in 10/24 patients; 6/14 patients with nasoseptal flap and 3/10 patients with TPFF were lost to follow-up after the first MRI.

### 3.2. Imaging Findings—Anatomical Spaces

Four anatomical spaces were evaluated for each of the 16 patients included in the final analysis. Inflammatory changes were observed in 17/64 spaces: the retropharyngeal space (9), parapharyngeal space (2), masticator space (5) and carotid space (1) (Table 1).

The regression of inflammatory alterations after 1 year was observed in 10/64 subsites: the retropharyngeal space (5), parapharyngeal space (1), masticator space (3) and carotid space (1).

Fibrosis was observed in five subsites in the first post-surgery imaging: the parapharyngeal space (2) and retropharyngeal space (3). No fibrotic changes were detected in the masticator and carotid spaces. Fibrosis was stable at 1-year follow-up.

Changes in the MRI findings between first and second MRI were not statistically significant (*p* > 0.05).

Pictorial representation of the results shown in Figure 1.

In Figure 2, an example of the expected findings in patients who underwent NER 3 is reported.

### 3.3. Skull Base Reconstruction

Twenty-four out of thirty-two patients with available imaging required reconstruction with a pedicle flap. The nasoseptal flap was used for defect coverage in 14 patients. TPFF was used in 10 patients.

The statistical analysis did not reveal any association between the type of flap used and the presence of either fibrosis or inflammation.

#### 3.3.1. Nasoseptal Flap

In the first MRI, the nasoseptal flap showed a mean thickness of 5.8 mm (3–11 mm) and showed high and homogeneous enhancement in most cases (13/14, 93%). As shown in Figure 3, enhancement and flap thickness were unchanged at the 1-year follow-up (*p* > 0.05).

#### 3.3.2. Temporo-Parietal Fascia Flap (TPFF)

The imaging findings of the three layers of TPFF (Figure 4) are reported in Table 2, and the signal changes at one year are listed in Table 3.

The main changes reported at 1 year for each layer are as follows:The mucosal layer showed a decrease in T2-signal intensity in more than half of the cases (four out of seven patients, 57%) and a reduction in enhancement in three cases (43%) at one year. In six out of seven patients (86%) the mucosal layer showed a thickness reduction one year after NER.The intermediate layer showed a decrease in the T2 signal in 86% of patients, while the T1 signal and enhancement were unchanged in the majority of patients (71%).The deep layer showed a T1 signal increase in five out of seven patients (71%), as shown in Figure 5.

## 4. Discussion

Imaging findings after NER have not been widely described; the few studies reported in the literature describe mainly MRI signal changes in skull base bones [21,22]. To our knowledge, this is the first study to describe the imaging features of deep spaces of the neck after endoscopic surgery. Reporting and describing these “expected findings” can be useful to radiologists to discriminate normal signals after NER from abnormal signals that might herald the presence of complications or recurrence.

In our study, signal changes were more observed in anatomic sites closer to surgical boundaries, and this is in accordance with the literature [21,22,23].

We found that the clivus and base of the pterygoid process are the skull base structures that more commonly display bone marrow edema; the petrous apex and atlo-occipital junction are less commonly involved (both in 6% of patients). At 1 year, a resolution was observed in half of the cases. Direct damage to the periosteum and the drilling of bone structures (in particular, the clivus) can explain bone inflammation. Sclerosis was rare and was found in only one case, in the clivus.

Similarly, soft-tissue spaces of the neck adjacent to the surgical boundaries also commonly present signal changes. The space more commonly involved was the retropharyngeal/prevertebral space: 56% of patients at 6 months presented inflammatory changes, which significantly decreased at 1 year. This space was also the most frequently involved in fibrosis, which remained stable in the 1-year follow-up. Inflammatory changes were also frequently reported at 6 months in the masticator space (30% of) patients; these changes were less common at 1 year and were identified in only 12% of patients. Both inflammation and fibrosis were found in 12.5% of patients in the parapharyngeal space at 6 months.

Inflammation can be expected in deep spaces of the suprahyoid neck as a response to direct trauma during the surgical procedure, or due to the passage of infective agents from the non-sterile nasopharyngeal cavity, in the post-surgical period and during the healing phase.

While inflammation regressed at 1 year in 60% of patients, fibrosis remained stable.

This study demonstrates that inflammatory changes might be a common finding, especially in areas more directly involved in the surgical treatment, such as the retropharyngeal space and the clivus; usually, these changes are nearly halved at one year and are expected to decrease progressively at subsequent follow-ups.

Our analysis of normal MRI findings after NER also focused on the MRI appearance of the reconstruction flaps, which has not been described so far in the literature. In particular, we report a detailed description of the tri-layer structure of the TPFF.

A normal nasoseptal flap appears on imaging as a thin layer of tissue at the level of the postero-lateral surgical border with homogeneous enhancement (93%), and its aspect is similar to the superficial mucosal layer of the TPFF. No significant changes over time should be expected in the nasoseptal flap apart from a minimal thickness reduction.

The TPFF typically displays a reduction in the T2 signal of the intermediate layer over time (probably due to edema reduction) and a slight reduction in enhancement at 1 year.

Interestingly, the deep layer of the TPFF is characterized by an overall high T2 signal at 6 months (70%), which is quite stable 1 after surgery; conversely, the T1 signal increases in the majority of patients. This combination of information for the two sequences suggests a resolution of inflammatory changes and normalization of the fatty signal typical of this layer of the flap.

This study has limitations, including the retrospective design, the low sample size that did not allow for statistical analysis, and the image analysis performed by a single reader, which could introduce interpretation bias given that the features we considered might be influenced by a degree of subjectivity.

## 5. Conclusions

Overall, follow-up MRI scans of patients treated with NER and reconstruction flaps of the skull base are infrequent occurrences in routine practice, due to the highly selective indications of this second-line treatment. Awareness of normal and expected postoperative findings is an essential prerequisite for the correct identification of complications and recurrence.

## Figures and Tables

**Figure 1 jcm-13-02624-f001:**
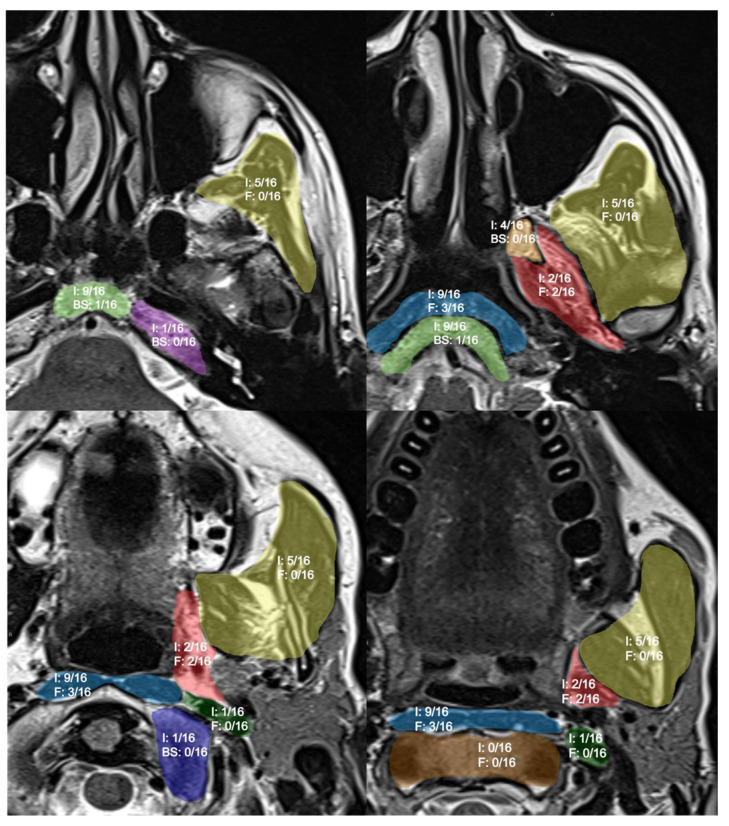
Pictorial representation of findings after NER. I: inflammatory changes; F: fibrosis; BS: bone sclerosis. Anatomical subsites are highlighted with different colors: masticator space in yellow; prevertebral space in light blue; carotid space in dark green; parapharyngeal space in red. Skull base subsites highlighted: clivus in light green; the base of pterygoid processes in orange; atlo-occipital joint in blue; petrous apex in purple; second cervical vertebrae in brown.

**Figure 2 jcm-13-02624-f002:**
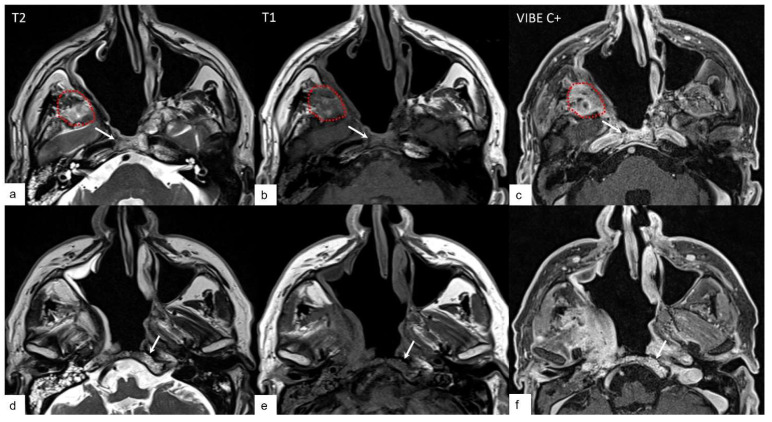
Expected findings in deep neck spaces and skull base bones: inflammatory changes. An example of bone edema in the clivus is denoted by the white arrow highlighting hyperintense T2 signal (**a**,**d**), hypointense T1 signal (**b**,**e**) and enhancement after contrast medium injection (**c**,**f**) in the bone marrow of the clivus; the red dotted line is encompassing an inflammatory change in the masticator space (hyperintense in T2-weighted sequence, hypointense in T1-weighted sequence and enhancement after contrast agent).

**Figure 3 jcm-13-02624-f003:**
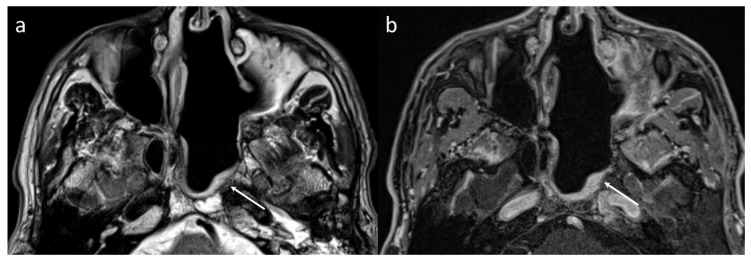
MR findings at the first MRI (6 months) of a nasoseptal flap showing high T2 signal in the T2-weighted sequence (**a**) and homogeneous enhancement in a 3D T1-weighted isotropic fat-saturated sequence acquired after contrast medium injection (**b**).

**Figure 4 jcm-13-02624-f004:**
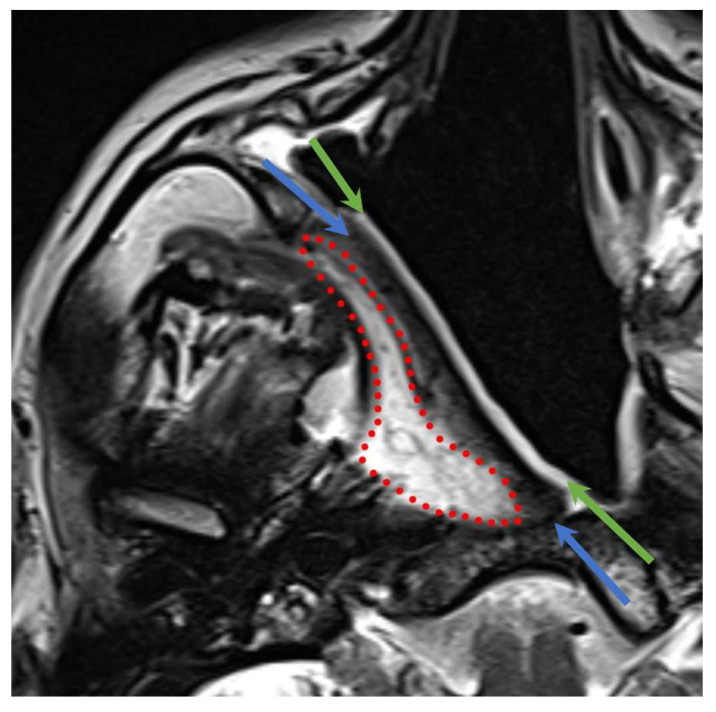
Multilayered aspect of the TPFF on T2-weighted sequence; red dotted line: deep layer; light blue arrows: intermediate layer; green arrows: mucosal layer.

**Figure 5 jcm-13-02624-f005:**
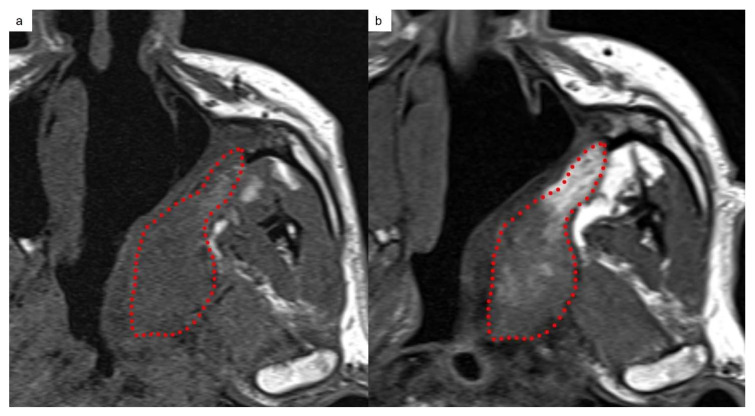
TPFF MRI appearance in a T1-weighted sequence 6 months (**a**) and 1 year (**b**) after NER showing a T1 signal increase in the deep layer (encompassed by the dotted line) of the flap at the second follow-up.

**Table 1 jcm-13-02624-t001:** Imaging findings in predetermined anatomical subsites, with findings of MRI at 6 months and 1 year after surgery.

Anatomical Subsite	Inflammation			Fibrosis		
	First MRI	Second MRI	Variation	First MRI	Second MRI	Variation
Parapharyngeal space	2	1	↓1	2	2	=
Retropharyngeal space	9	4	↓5	3	3	=
Masticator space	5	2	↓3	0	0	=
Carotid space	1	0	↓1	0	0	=

**Table 2 jcm-13-02624-t002:** Normal TPFF appearance on 6-month follow-up scan.

First MRI (*n* = 10)	T2 Signal	T1 Signal	Enhancement
Mucosal layer	9 hyperintense	10 hypointense	9 high
	1 hyper/isointense		1 mild
Intermediate layer	5 hypointense	10 isointense	2 high
	4 isointense		8 mild
	1 hyperintense		
Deep layer	7 hyperintense	9 isointense	5 high
	1 hyper/hypointense	1 hyperintense	5 mild
	2 isointense		

**Table 3 jcm-13-02624-t003:** Normal TPFF appearance on 1-year follow-up scan.

Second MRI (*n* = 7)		T2	T1	Contrast Enhancement	Thickness (mm)
Mucosal layer	Decreased	4	-	3	6
	Increased	-	-	-	-
	Unchanged	3	7	4	1
Intermediate layer	Decreased	6 **	-	2	1
	Increased	-	-	-	1
	Unchanged	1 **	7	5	5
Deep layer	Decreased	1	-	3	1
	Increased	-	5 *	1	-
	Unchanged	6	2 *	3	6

*: *p* = 0.034; **: *p* = 0.055.

## Data Availability

The datasets analyzed during the current study are available from the corresponding author upon reasonable request.

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
