# Peer review of "Magnetic Resonance Imaging after Nasopharyngeal Endoscopic Resection and Skull Base Reconstruction"

_jcm, 2024, doi:10.3390/jcm13092624_

Round 1
Reviewer 1 Report
Comments and Suggestions for Authors
I would like to congratulate authors for conducting such a qulified study among postoperative patients repaired with flaps in this very selected group of patients. This study will help for better distinguishing between normal and expected postoperative findings among patients treated with surgery.
Author Response
Dear colleague,
many thanks. We are grateful for your positive comments.
The intent of the paper was actually to describe what radiologists or surgeons must address when looking at MR studies after such complex surgical procedures.
Kind regards
the authors
Reviewer 2 Report
Comments and Suggestions for Authors
I want to thank the Authors for all their work, it will be a beneficial addition to the literature. Overall, the article provides very useful information. However the two types of flap repair described in this article vary significantly in technique and invasiveness, and the repair it self may have an effect on both fibrosis and inflammation in post operative scans. Can the Authors comment on how the type of repair effected the post op imaging in the various locations evaluated?
Also there are multiple grammatical mistakes that add a degree of confusion including in lines 180-81 the way it is written is unclear it might be better to say 4 subsites were evaluated in 16 people
I would also use the same format for table 3 and table 4 as it is a little distracting.
Comments on the Quality of English LanguageThere are multiple grammar errors as well as missing words
Author Response
We are grateful for your appreciation and your kind consideration.
Basing on your comment and suggestion, we have performed a Fisher's exact test which demonstrated tha absence of significant association between the kind of harvested flap and the presence of scar, edema or granulation tissue.
We have rephrased lines 180-181 to improve the clarity of the sentence as you suggested.
As you pointed out we have made some changes to table 3 and table 4 to help the readers to understand the data reported.
An English editing by a native language speaker has been performed and changes to improve English throughout the article have been made.
Reviewer 3 Report
Comments and Suggestions for Authors
The authors studied the expected findings on MRI after nasopharyngeal endoscopic resection and skull base reconstruction. The findings has been well described. However, I believed the most important role of MRI in such of cases is to differentiate between the non pathological findings and the recurrence which was not addressed in this study. Therefore, I think the manuscript does not add much to the existing knowledge.
Author Response
Dear colleague,
thanks for reviewing our paper and for providing your valuable comments.
You are perfectly right when you assert that the main aim of radiology is to differentiate expected from pathological findings after NER.
We have already analyzed the pathological conditions of recurrences and complications but we decided to keep this material for a second paper, in order to not overload the present study which, in our opinion, already contains much information. Combining the two papers we could provide a comprehensive scenario of imaging after NER.
Thanks again.
Kind regards,
the authors
Reviewer 4 Report
Comments and Suggestions for Authors
The authors have presented a study of 16 patients who underwent endoscopic surgery by assessing their respective expected MRI findings postoperatively up to a year. Although the study sounds appealing, the study has flaws which need to be adressed.
Although it is appreciated that these procedures are not commonplace in all tertiary centres, the number of patients included is quite significantly low. Over a decade, a larger cohort would be expected. It would perhaps be prudent to present this as a case series.
The above point leads to the next issue, which is the lack of statistical analyses to determine if these findings after 6 and 12 months are clinically signficant. It is difficult to understand whether these findings would be useful to clinicians.
The MRIs have been evaluated by only one radiologist, which inherently introduced observer bias. It would be better to include at least 2 radiologists and discuss inter-observer compliance of the MRI findings.
Under section 2.6 Image analysis, the authors have mentioned that bone changes were evaluated as well, which could be better done with CT scans rather than MRI, although sclerotic changes to bone is appreciably evaluated with MRI because of higher sensitivity.
Although perhaps interesting to a select group of clinicians (radiologists), I cannot see the major utility of this in clinical practice with respect to guiding further management of this patient group.
Author Response
Dear colleague,
many thanks for reviewing our paper and for giving your valuable inputs.
As you commented, the number of cases is very low compared to the number of surgical procedures (50) performed in our center. This is due to two reasons:
- exclusion of cases without appropriate imaging studies
- exclusion of cases with complications or recurrence (the aim of this study is actually to describe the expected findings, while complications and recurrences will be addressed in a second paper).
You are completely right when you underline the absence of statistical analysis as an important limitation of our study.
Also, the absence of a second reader is a weakness of this study and it will be mentioned among the limitations.
When considering bone changes, is true that CT is better than MRI in evaluating sclerotic changes, but is not able to identify bone marrow edema and enhancement: the multiparametric evaluation approach of MRI is preferrable for these reasons in this setting.
We hope to have appropriately answered your questions.
Many thanks again.
Sincerely yours,
the authors
Reviewer 5 Report
Comments and Suggestions for Authors
JCM (ISSN 2077-0383)
Manuscript ID jcm-2900069
Magnetic Resonance Imaging after Nasopharyngeal Endoscopic Resection (NER) and Skull Base reconstruction
The paper is well-written and addresses a significant gap in current research.
The authors conducted a comprehensive study on the imaging findings after Nasopharyngeal Endoscopic Resection (NER) and skull base reconstruction. The detailed description of expected findings in both the anatomical spaces and reconstruction flaps provides valuable insights for radiologists involved in post-operative imaging interpretation.
There seems to be a scarcity of research specifically focusing on MRI findings after Nasopharyngeal Endoscopic Resection (NER) itself, with most studies concentrating on skull base changes or recurrent NPC (1). It is challenging to distinguish between normal post-operative changes and signs of a tumor recurrence. The paper by Li et al. (2013) (2) seems to be one of the few that directly explores MRI changes in the skull base bone after NERS for recurrent NPC. Many studies delve into MRI findings after various skull base surgeries, including those with reconstruction techniques, but these studies do not associate the specific effects of NERS.
This study directly tackles the challenge of differentiating normal post-operative findings after NERS using MRI. It provides a detailed analysis of signal changes in both skull base structures and deep neck spaces. The inclusion of a detailed description of the MRI appearance of reconstruction flaps (nasoseptal and temporoparietal fascial flaps) is valuable for radiologists. Overall, the paper contributes valuable knowledge to the field and has the potential to improve post-operative MRI interpretations for NERS patients.
Minor suggested changes:
Line 189: In the legend of table 1 you can add “findings at MRI at 6 months and 1 year after surgery” and in the table 1 instead of : First MRI Second MRI you could use MRI 6 m and MRI 12 m
(1) Teo, P.T.H.; Tan, N.C.; Khoo, J.B.K. (2013). Imaging appearances for recurrent nasopharyngeal carcinoma and post-salvage nasopharyngectomy. Clinical Radiology, 68(11), e629–e638. doi:10.1016/j.crad.2013.06.003
(2) Li H, Wang DL, Liu XW, Chen MY, Mo YX, Geng ZJ, Xie CM. MRI signal changes in the skull base bone after endoscopic nasopharyngectomy for recurrent NPC: a serial study of 9 patients. Eur J Radiol. 2013 Feb;82(2):309-15. doi: 10.1016/j.ejrad.2012.10.022. Epub 2012 Nov 21. PMID: 23177186.
Author Response
Dear collegue, thank you. We are really glad that you appreciated our effort of giving a comprehensive knowledge of expected findings after NER and skull base reconstruction. A second paper focused on imaging of recurrences and complication will follow.
We have followed your suggestion and changed the Table 1 legend as you requested, we have maintained First MRI and Second MRI as column headers to maintain consistency with Tables 2 and 3.
Kind regards,
the authors.
Round 2
Reviewer 3 Report
Comments and Suggestions for Authors
The authors studied the findings on MRI after nasopharyngeal endoscopic resection and skull base reconstruction. It was an interesting study and important clinically. The article has been satisfactorily revised.
Author Response
Dear colleague,
Thank you for your positive comment,
Kindest regards,
the authors
Reviewer 4 Report
Comments and Suggestions for Authors
It's unfortunate that essentially no signficant changes were done to the manuscript. At the very least, it would perhaps be clinically useful to have non parametric statistical analyses of the limited cohort as this could guide further research forwards.
Comments on the Quality of English LanguageThe English quality has been improved since first revision,, which seems to be satisfactory now
Author Response
Dear Reviewer,
Thank you for your continued evaluation of our manuscript and for your insightful suggestion regarding the inclusion of statistical analyses. We have acted upon your recommendation and performed these analyses on our limited cohort.
The results of this additional analysis seem to be interesting. We found that the changes in the deep layer of the Temporoparietal Fascia Flap (TPFF) were statistically significant (increase in T1 signal). Additionally, our findings indicate that changes in the intermediate layer of the TPFF also exhibit a tendency toward statistical significance (decrease in T2 signal).
Another result higlighted by statistical analysis is that there isn't any relation between the type of flap used and the presence of fibrosis or inflammation.
These findings have been integrated into the revised manuscript. We believe that this new data not only addresses your initial concerns but also adds substantial depth to our study, enhancing its relevance and potential to guide future clinical research.
Once again, we appreciate your constructive feedback and hope that our revisions now meet the standards of the journal.
Sincerely,
the authors